# Evaluating a Novel Extended Scope of Occupational Therapy Service Aimed at Hospital Avoidance in Tasmania, Australia, from the Perspective of Stakeholders

**DOI:** 10.3390/healthcare10050842

**Published:** 2022-05-03

**Authors:** Pieter Jan Van Dam, Leah Reid, Sarah Elliott, Mitchell Dwyer

**Affiliations:** 1School of Nursing, College of Health and Medicine, University of Tasmania, 4–8 Bass Hwy, Burnie, TAS 7320, Australia; pieter.vandam@utas.edu.au; 2Royal Hobart Hospital, Tasmanian Health Service, 48 Liverpool St., Hobart, TAS 7000, Australia; sarah.elliott@ths.tas.gov.au; 3Tasmanian School of Medicine, College of Health and Medicine, University of Tasmania, Medical Sciences Precinct, 17 Liverpool St., Hobart, TAS 7000, Australia; mitchell.dwyer@utas.edu.au

**Keywords:** hospital avoidance, occupational therapy, environmental assessment

## Abstract

The Australian state of Tasmania has seen a spike in Emergency Department presentations in recent years, particularly among the elderly. A novel extended scope occupational therapy (ESOT) service was implemented by the Tasmanian Health Service, aimed at supporting hospital avoidance. Clients were referred to the ESOT service by other services after being assessed as having a high risk of imminent hospital presentation. Occupational Therapists provided short-term interventions related to falls, mobility, nutrition, and initiated onward referrals to other services. A convergent parallel mixed methods design was used to evaluate the ESOT service. Quantitative data from routinely collected administrative records and a purpose-built survey of referring clinicians were used alongside qualitative data from semi-structured interviews with clients/carers, to enable the triangulation of data. Quantitative data were analysed using descriptive statistics, while qualitative data collected in interviews were thematically analysed. A total of 104 extended scope interventions were provided to 100 clients. Most clients were able to stay at home. Qualitative data revealed that mobility, support, and facilitating access to support services were factors which added value to the client and carer experience. In conclusion, the ESOT program contributed to potentially avoiding hospital admissions and to improving the quality of life of participating clients.

## 1. Introduction

Australia’s public health system is at present characterised by rising costs, workforce shortages and prolonged waiting periods, among other factors [1]. These pressures are particularly evident in the state of Tasmania, which has seen a significant increase in the number of emergency department presentations during recent times [2]. Tasmania has the highest median age of all Australian states and territories [3]; it is therefore reasonable to expect that a large portion of its emergency department presentations relate to falls, which are a leading cause of presentations in Australians aged over 65 [4]. The sequelae of falls are notable, with evidence suggesting that 13–33% of individuals who present to emergency departments for a fall will experience a subsequent fall within six months [5,6], and 46–52% will fall again within twelve months [7,8]. Indeed, in 2015–2016, falls accounted for $3.7 billion of health expenditure nationally, making it the condition with the second highest expenditure [9]. These circumstances call for a rethink of traditional modes of service delivery, which do not appear to be meeting the needs of clients at risk of falls.

One novel approach is the use of the extended scope of practice services, which are broadly defined as where clinicians work “beyond the recognised scope of practice of the profession of interest in innovative or non-traditional roles” [10]. The use of such services has been adopted in various allied health professions in numerous different forms [10,11]. Taken together, these findings and several others [12] highlight the value of tailored, multidisciplinary services working across the continuum of hospital and community to address clients’ difficulties with activities of daily living. Occupational Therapists are well-positioned to contribute to such a service. Despite this, there have been few Australian studies on the use of extended scope of practice Occupational Therapists in community settings [13,14,15] and fewer still which have captured the clients’ perspective [16,17]. With a greater understanding of how the skillset of Occupational Therapists could be complemented by further training in related disciplines, clinicians and policymakers would be better able to develop models of care which utilise these skills to reduce the risk of falls in community-dwelling adults.

The Tasmanian Health Service funded a 12-month trial of an extended scope occupational therapy (ESOT) outreach position to provide a rapid response 7-day service to people living in the community who were identified by other services (e.g., Ambulance Tasmania, Community Nursing, General Practitioners, and the emergency department-based Emergency Multidisciplinary Assessment Team (EMAT)) as having a high risk of imminent hospital presentation. The ESOT service was based from the Royal Hobart Hospital, the largest tertiary care centre in the state, with an estimated catchment population of approximately 250,000 [18].

The ESOT service used a transdisciplinary approach to assess and provide short-term interventions for people, whereby Occupational Therapists working in the service received additional competency training in the areas of nutrition and mobility assessment. Training in nutrition was delivered by a community-based dietetics service in two one-hour long online training modules, and covered malnutrition risks, screens/assessments, and nutrition intervention options. Training in mobility was in the form of a one-day session delivered by a Senior Community Physiotherapist from the Tasmanian Health Service and focused on stair assessment, balance assessment, basic gait assessment/gait aid prescription, and education of weight bearing status. Each ESOT staff member also attended one visit to a client’s home where they could observe a community physiotherapist assessing a client.

ESOT assessments took place in the client’s home and included an evaluation of their cognition, mobility, transfers, nutrition, social supports, and the overall environment. Interventions provided to clients included both regular ‘core’ Occupational Therapist duties and extended scope capabilities. Extended scope interventions related to either clients’ mobility, nutrition, or facilitating access to support services (e.g., My Aged Care, respite care). Core duties were grouped into one of the below categories: Equipment and small aids (e.g., to assist with activities of daily living (ADLs); Home modifications (e.g., installation of grabrails); Falls education; Occupation changes (e.g., drying laundry on a rack inside, as opposed to on a clothesline outside); Person-related changes (e.g., sitting as opposed to standing while getting dressed); Cognition (e.g., memory-related strategies); Pressure injury-related; Onward referrals (e.g., physiotherapy, dietetics, and general practitioners).

Onward referrals were provided to other services where follow up was required. To further explore the potential of extended scope Occupational Therapy services and their utility from the perspective of clients/caregivers and referring clinicians, a mixed methods evaluation of this pilot service was undertaken. It is also hoped that the evaluation will inform future decisions on whether to continue the service; and if so, to determine what improvements could be made; or if not, which alternate models of care may be more cost effective and appropriate. The aim of this study was to explore the potential of ESOT services regarding hospital avoidance and the Service, and how they are perceived by clients/caregivers and referring clinicians.

## 2. Materials and Methods

A convergent parallel mixed methods design as described by Creswell [19] was used to evaluate the ESOT service. This design was chosen to enable triangulation of data from different perspectives and to validate the findings generated by each method through evidence produced by the other [19,20]. Two sources of quantitative data, namely (i) routinely collected organisational data and (ii) a purpose-built survey of referring clinicians, were used alongside qualitative data from semi-structured interviews with clients and their carers.

### 2.1. Characteristics and Recruitment of Participants

Consenting clients were referred to the ESOT service due to difficulties in activities of daily livings, mobility, cognition, social changes (e.g., carer illness) or due to a risk of falls, pressure injury, or malnutrition. The inclusion criteria for ESOT service were people deemed to be at risk of presentation to the emergency department within the next 5 days, who resided within approximately a 1-h drive of the base. Exclusion criteria were those deemed medically unstable, requiring immediate presentation to emergency department, or aged under 18 years. Clients with acute substance, psychiatric, gynaecological, or obstetric conditions were out of scope from the ESOT service.

A consecutive sample of the first 140 individuals to be referred to the ESOT service was screened for eligibility for inclusion in the study; this number was representative of a much larger number of clients seen by the ESOT service during the trial period.

Purposive sampling was used to recruit the following two participant groups.

Clients and carers who accessed the service were recruited in their home by ESOT Occupational therapists towards the end of their episode of care with the ESOT service. Written consent was obtained for a face-to-face interview to be carried out in their own home by a non-therapist member of the research team.

Referrers to ESOT were recruited by email to participate in an online survey. The email was sent to key stakeholders of each organisation, with a request that it be passed on to their staff. The email contained a link to an online survey (via Survey Monkey) which included a statement explaining the purpose of the survey and consent to participate.

### 2.2. Data Collection

Routinely collected organisational data were obtained by reviewing the clinical records of clients who had received the ESOT service and manually abstracting data from ESOT assessment forms, which were completed by Occupational therapists for all clients who received an ESOT service. This included clients’ demographic data (e.g., age, gender, residential status), the number of reasons for their referral (i.e., as a proxy for patient complexity), and the number of recorded falls within the past 12 months. The same assessment form contained clients’ risk of falls, malnutrition, and pressure injury. Falls risk was calculated using the validated Falls Risk for Older People in the Community (FROP-Com) tool [21,22], whereas the Braden Scale [23] and Malnutrition Screening Tool (MST) [24] were used to assess clients’ risk of pressure injury and malnutrition, respectively. Clients’ socio-economic status was calculated using the Index of Relative Socio-economic Disadvantage (IRSD) [25].

Data pertaining to ESOT service delivery were also captured in this form. This included timestamps for the referral from other services and the date clients were seen by the ESOT service, along with their referral priority category. These were determined by the ESOT clinicians who received and triaged referrals and were grouped into three tiers:Priority 1 = urgent, to be responded to within 24 hPriority 2 = semi-urgent, to be responded to within 3 daysPriority 3 = less urgent, to be responded to within 5 days

Lastly, details of the interventions provided to clients, along with the immediate outcome of the ESOT service (i.e., in terms of advocated and prevented admissions and changes in unmet activities of daily living needs) were also obtained from ESOT assessment forms. Qualitative data were collected through use of semi-structured face-to-face interviews with consenting clients and/or caregivers who had accessed the service. All interviews were conducted in clients’ homes by one member of the research team who was not directly involved with the ESOT service, and were audio recorded for transcription and analysis. Data collection for all sources occurred concurrently during the period of 2 January 2018 and 30 November 2019. Ethical approval was obtained from the University of Tasmania’s Human Research Ethics Committee (Application H0017864).

### 2.3. Data Analysis

Descriptive statistics were used to analyse the quantitative data. Interview data were analysed via thematic analysis using the Thematic Network technique described by Attride–Stirling [26]. This method involves coding line by line to develop Basic Themes followed by the organisation of data into Organising Themes. These themes are then further categorised into overarching, Global Themes that encompass the principal metaphors in the data as a whole. In this study, global themes represent the position of the participants about their experience with the ESOT service. The data derived from three sources was triangulated to provide a holistic view on the service provided by ESOT.

## 3. Results

### 3.1. Quantitative Findings

Of the initial 140 subjects, 40 did not have home assessments, either because they were found to be ineligible for the service, or they declined their referral. Figure 1 illustrates the flow of participants through the study. Table 1 (below) shows the participants’ baseline characteristics. Over 44% of clients had experienced at least one fall in the preceding twelve months, and an overwhelming majority (79%) had more than one reason for being referred to the ESOT service. Applying the IRSD [24] to the cohort under investigation showed that most individuals n = 54 (38.6%) fell in the most disadvantaged quintile, while the lowest number n = 27 (19.3%) of individuals fell in the most advantaged quintile.

As shown in Figure 1 below, following assessment by the team, the most common interventions delivered to clients were in the form of falls education, equipment/small aids (e.g., grab rails, walking aids and bathing equipment) and occupation changes. Only six clients (6%) did not have any interventions provided to them.

Table 2 shows clients’ risk of pressure injury, malnutrition, and falls following their ESOT assessment. Almost half (45%) of ESOT clients were found to be at a high risk of falling based on their FROM-Com category, whereas scores on the MST and Braden Scale were comparatively mild.

Table 3 shows the number of individuals from each of the three referral categories who were responded to within the timeframe specified by their referral category. All but two clients in the highest priority category (referral category 1) were responded to in a timely manner.

Table 4 shows the outcomes of clients assessed and treated by the ESOT service from a hospital avoidance perspective. A clear majority of clients (81%) were deemed to have had a significant risk addressed by the ESOT team and likely prevented a hospital admission. A smaller number of clients (6%) were recommended for hospital admission, and the remainder were either in need of a service not provided by the team (7%) or had an undetermined outcome (6%).

Table 5 shows the number of interventions provided to clients which were considered part of the core Occupational Therapy service delivery and those which utilised clinicians’ extended scope capabilities in mobility and nutrition. Although core Occupational Therapy interventions comprised the majority of interventions, a substantial number of extended scope interventions were provided.

The survey of referring clinicians returned 30 responses, of which 16 (53%) were from community or district nursing staff, 6 (20%) Ambulance Tasmania staff, and the remainder from the EMAT, community allied health, or other groups. Table 6 below provides a summary of the responses. At least three quarters of respondents either agreed or strongly agreed to statements supporting the timeliness, utility, and ease of use of the ESOT service. Though only one negative response about the service was given, several respondents were unsure about the value of the service to clients.

### 3.2. Qualitative Findings

Ten interviews with clients (n = 4) and carers (n = 6) were conducted from May to August 2019, all of whom had engaged with ESOT on at least one occasion previously. Participants’ ages ranged between 41 and 86 years. All participants resided in the greater Hobart region. The analysis of the interview transcripts using Attride-Stirling Analytic Tool for Qualitative Research [25] generated three global themes, providing a deeper understanding of the experience of clients and carers using ESOT. Figure 2 below shows the key features of the analysis depicted as a thematic network [26]. Data sufficiency was reached after 10 interviews, as no new insights were uncovered.

#### 3.2.1. Global Theme 1: Staying at Home Is Preferred

The importance of staying at home was expressed by all participants, and many participants did not like to visit hospitals. Some participants reported feeling scared of being in a clinical environment, whereas others feared that once admitted to hospital, they would not be able to return home. Some participants recalled stories in which friends and relatives of theirs were transferred to a nursing home after having been admitted to hospital.


*“He doesn’t want to leave home, it is his world.”*


#### 3.2.2. Global Theme 2: Preventing Falls through Tools and Education

Participants believed that the tools and education provided to them prevented falls. Including partners when being educated on the use of these tools was regarded as essential, as it provided reassurance that the likelihood of a fall was significantly reduced.


*“I don’t have to watch him all the time now.”*


The educational component allowed participants to feel more confident in using the tools, and many expressed that what was taught was easy to apply in their home environment. The application of the learning was seen by many participants as creating a sense of independence, as the use of the tools helped with confidence in participating in activities of daily livings.

#### 3.2.3. Global Theme 3: Feeling Important

Some participants received multiple visits from the ESOT staff, and the time spent with clinicians made them feel important. Moreover, participants expressed that the way the staff of the service listened and explained care requirements made them feel ‘human’.


*“She had time for him and explained it to him in such a way that he understood.”*


Many participants described their experiences of navigating the healthcare system as ‘challenging’, feeling unsure about what service they required and how to access it. Furthermore, they felt overwhelmed with the high number of different healthcare providers.


*“If you have someone sick (in your family) you don’t know where to get help.”*


The advice and arrangements the ESOT helped to put in place made them feel relieved and reduced their anxiety about arranging care.

## 4. Discussion

The aim of this study was to explore the potential of ESOT services and how they are perceived by clients/caregivers and referring clinicians. Triangulation of the data showed that the service has delivered value to all stakeholders. It is clear that the service has provided a holistic approach towards the care of referred clients, leading to better client outcomes by preventing imminent risks and both preventing and advocating admissions as required. ESOT interventions relating to mobility and facilitating access to support services appeared to have played a large role in achieving these outcomes.

The primary aim of the ESOT service is hospital avoidance, which was largely achieved. This was evident in the service’s prompt response times for referrals, particularly for urgent cases. More pertinently, over 80% of clients were deemed to have avoided hospital admission or had a significant risk addressed as a result of being seen by ESOT clinicians. This was validated in survey responses from referring clinicians, most of whom agreed that the ESOT service supported their clients to avoid a hospital admission. Moreover, as demonstrated in our qualitative data, by allowing clients to remain in their homes, the ESOT service was able to allay clients’ concerns about being taken away from their home indefinitely and placed in a nursing home.

Through ESOT assessment and intervention, the aim was to help to reduce falls, improve activities of daily living unmet need and enable faster access to health services. ESOT utilised an in-depth assessment including determining the risk of pressure injuries using the Braden Scale, risk of malnutrition using the MST, and falls risk using the FROP-Com tool. Many of these tools would typically be used by multiple other healthcare professionals, however, under the ESOT service, one clinician can do all of the above, negating the need for clients to interact with numerous clinicians and repeat their history to each one. This not only demonstrates continuity of care [27], but also a streamlined process [28].

A large number of study participants had a high baseline risk of falling, and the majority of interventions provided were in some way related to falls prevention. Taken together, these findings suggest that clients in this category derived the most benefit from the ESOT service, and therefore likely represented most cases of hospital avoidance. This notion is supported by the qualitative data taken from clients and their carers, who spoke of the benefit of tools/equipment and education, and the resulting impact it had on their risk of falling. The benefit of using community-based Occupational Therapists to reduce the risk of falls is well established [12,29]. At the same time, the number of participants with a high baseline risk of malnutrition, and hence the number of nutrition-related interventions provided, was comparatively small. This is likely a reflection of a relatively low prevalence rate of malnutrition among community-dwelling elderly people, when compared to those living in long-term care [30].

The number of referrals to other healthcare professionals demonstrated that care facilitation was required in the community setting. Our qualitative data suggested that this was value-adding from the client perspective, with clients recalling previous negative experiences of navigating the health system, and reporting a sense of relief that their care was being arranged. This was further corroborated in survey responses from referring clinicians, where most respondents agreed that the ESOT service supported comprehensive care of the clients they referred. Care facilitation is regarded as a vital component of patient-centred care [31] and care facilitation in the community is often regarded as fragmented [32]. The ESOT service established an important facilitation role which was previously lacking within in the primary health care setting, thus contributing to the wellbeing of clients and carers. This finding, along with the endorsement of referring clinicians, provides support for continuation of the ESOT service beyond its initial trial period.

It was observed that a large proportion of clients (38.6%) of the ESOT belonged to the most economically and socially disadvantaged cohort in the Australian society. This in a way was not a surprising finding, as socioeconomic factors are important determinants of health. People living in lower socioeconomic areas are regarded to be at greater risk of developing health issues, have higher rates of illness and disability than people live in higher socioeconomic areas [33]. It was noted that many participants lived in houses not fit for purpose, with features such as outdoor areas, internal stairs and bathrooms without showers. Poor housing is associated with poor health, and improving housing conditions through ESOT’s mobility interventions could lead to improved health outcomes for clients [34].

This study has demonstrated that the extended scope of practice for Occupational Therapistss can build efficiency and effectiveness into the system. This is in line with studies investigating extended roles in other healthcare professions, such as nurse practitioners, advanced practice nurses, and advanced radiation therapists, which found that these roles enhanced quality of care, leading to better patient outcomes [35,36,37]. Yet, there have been very few studies which have reported on patient outcomes associated with extended OT roles, with most research in this area focusing on ailment-specific programs (e.g., arthritis care) [11]. Further parallels between the current study’s findings can be found internationally. Indeed, studies from the United Kingdom reported that Occupational Therapists working with paramedic services led to cost savings associated with reductions in fall rates, Emergency Department presentations and subsequent admissions [38,39]. Another British study evaluated a community-based occupational therapy service aimed at modifying tasks and environments for people with activities of daily living deficits over a four-year period, finding that 90% of clients experienced statistically significant increases in their performance and satisfaction with activities [40]. There are a few studies conducted in Scandinavia and Canada reporting on the positive impact of Occupational Therapist practices related to lifestyle interventions, health promotion, and prevention [41,42]. Our study complements these findings by providing support for the effectiveness of extended Occupational Therapist roles in an Australian context, and in doing so, has helped to secure ongoing funding for the ESOT service. Future research is needed to determine the ideal combination of extended scope capabilities for Occupational Therapists to possess when working with a given client population. As mentioned above, it is likely that some capabilities are seldom needed in daily practice. If this were to be the case, it would not be cost-effective to acquire and maintain such skills. Similarly, given the large number of onward referrals made to physiotherapy services, there is impetus to upskill in this area. It is also possible that some capabilities not trialed in this study (e.g., psychosocial screening using the Geriatric Depression Scale [43]) may have offered some additional benefit to clients, and these would be worth investigating.

This study contains some limitations which must be addressed. Firstly, the absence of a control group receiving core Occupational Therapist care makes it difficult to determine the additive effect of the Occupational Therapists’ extended scope capabilities on clients’ outcomes. Notwithstanding this, the number of extended scope interventions provided (a median of one per client) would suggest that an area of need was being met, and therefore, some protective effect was conferred to clients. Secondly, the Occupational Therapists in this study were not equipped with their extended scope abilities at the beginning of the study—these were gradually acquired over the course of the study period. For this reason, the number and breadth of extended scope interventions reported in this study (and, by extension, the corresponding client outcomes) may not be representative of those currently produced by the service. Thirdly, this study investigated addressing imminent risks contributing to reducing avoidable hospitalisations. However, the study design was limited in that it could not prove definite causation.

## 5. Conclusions

The evaluation of the ESOT services has shown that there is a need for the care provided by this service and that the service has made a positive impact on the lives of clients and their carers. The ESOT interventions have contributed to possible hospital avoidance, benefiting clients and healthcare organisations. This was evident through the triangulation of three sources of data. Considering extended scope roles for Occupational Therapists and alternative ways of delivering care in resource poor health services will help to create a more effective and efficient healthcare system. Care will need to be taken in ensuring that this cohort is well educated and trained before the service is rolled out. Future research is needed to determine the ideal combination of extended scope of Occupational Therapist services to provide care to client populations.

## Figures and Tables

**Figure 1 healthcare-10-00842-f001:**
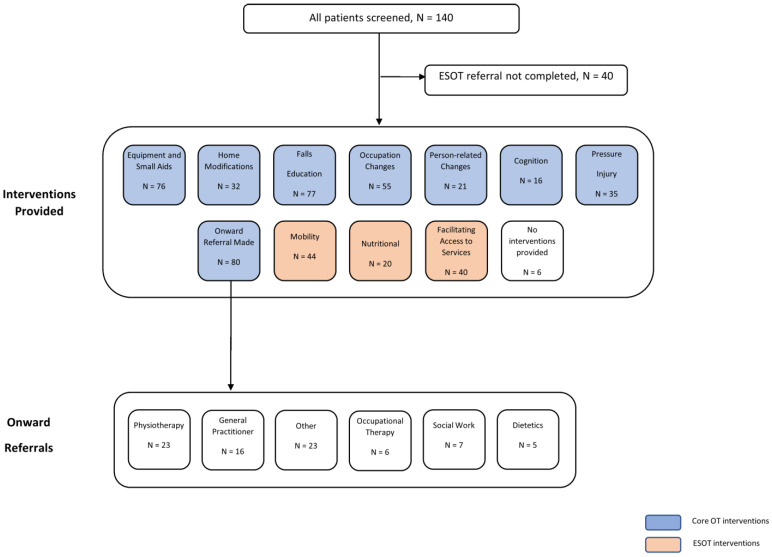
Flow of participants through study.

**Figure 2 healthcare-10-00842-f002:**
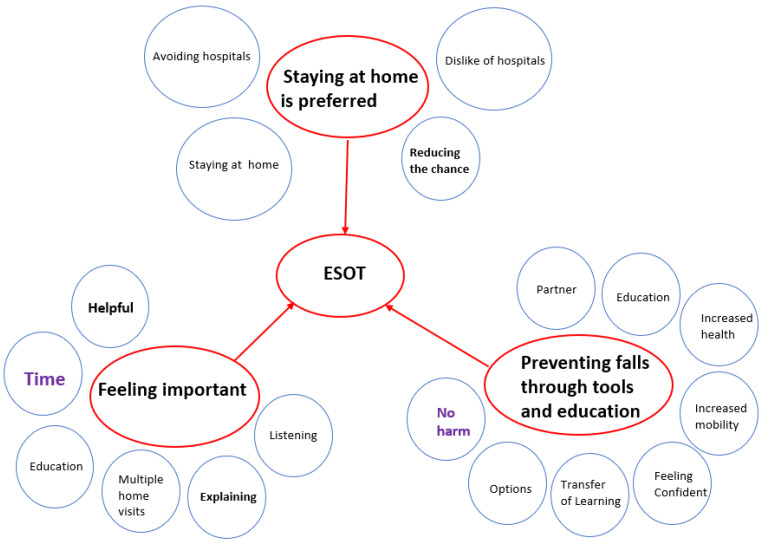
Thematic network.

**Table 1 healthcare-10-00842-t001:** Participant characteristics at baseline (n = 140).

	n	%
Age, mean (SD) (years)	76.8	(12.4)
Male	53	(37.9)
Female	87	(62.1)
Lived Alone	61	(43.6)
Index of Relative Socio-economic Disadvantage		
Quintile 1 (most disadvantaged)	54	38.6
Quintile 2	4	2.9
Quintile 3	27	19.3
Quintile 4	28	20.0
Quintile 5 (most advantaged)	27	19.3
Falls in past 12 months		
None	78	(55.7)
1	23	(16.4)
2	16	(11.4)
3+	23	(16.4)
No. Reasons for Referral ^†^		
1	24	(20.7)
2	44	(31.4)
3	32	(22.9)
4	24	(17.1)
5	7	(5)
6	4	(2.9)

^†^ Reasons for referral included: acute difficulty in coping, unmanaged acute cognitive decline, nutritional concerns, high falls risk/significant change in mobility, required increased support for acute change in function, social changes/needs impacting on ability to remain living at home, multiple falls in last 2 weeks or at risk of significant injury (if falls risk not addressed).

**Table 2 healthcare-10-00842-t002:** ESOT assessments of clients’ risk of pressure injury, malnutrition, and falls (n = 100).

Assessment	n	%
Braden Scale Score		
Moderate (13–14)	8	(8)
Mild (15–18)	24	(24)
None (19–23)	38	(38)
Not scored	30	(30)
Malnutrition Screening Tool Score		
High (3–5)	9	(9)
Moderate (2)	9	(9)
Low (0–1)	52	(52)
Not scored	30	(30)
Falls Risk for Older People in the Community Category		
High (4–9)	45	(45)
Low (0–3)	16	(16)
Not scored	39	(39)

**Table 3 healthcare-10-00842-t003:** No. of participants responded to within timeframe of their referral category (n = 100).

Referral Category	n	%
1 (within 24 h)	35/37	94.6%
2 (within 3 days)	37/49	75.5%
3 (within 5 days)	8/14	57.1%

**Table 4 healthcare-10-00842-t004:** Potential for hospital avoidance following ESOT assessment and interventions.

Outcome	n	%
Addressed significant risk/prevented imminent admission	81	81
Advocated admission	6	6
Requiring care not provided by ESOT service	7	7
Not stated	6	6
**Total**	**100**	**100**

**Table 5 healthcare-10-00842-t005:** Summary of interventions provided by type.

Intervention Type	Total Interventions Provided	Median No. of Interventions Per Subject
Core Occupational Therapist	392	4
Extended Scope Occupational Therapist	104	1
Core + ESOT overall	496	4

**Table 6 healthcare-10-00842-t006:** Summary of survey responses (n = 30).

Question	Strongly Agree	Agree	Disagree	Strongly Disagree	Do Not Know	Total
The service made it easy for me to refer to Allied Health	18 (60%)	7 (23.33%)	1 (3.33%)	0 (0%)	4 (13.33%)	30
I was able to get timely allied health services for the client I referred	20 (66.67%)	4 (13.33%)	0 (0%)	0 (0%)	6 (20%)	30
The service supported comprehensive care of the client/s I referred	20 (66.67%)	4 (13.33%)	0 (0%)	0 (0%)	6 (20%)	30
The service supported the people I referred to avoid emergency presentation or hospital admission	18 (60%)	5 (16.67%)	0 (0%)	0 (0%)	7 (23.33%)	30
The service should continue	24 (80%)	2 (6.67%)	0 (0%)	0 (0%)	4 (13.33%)	30

## Data Availability

The data presented in this study are available on request from the corresponding author. The data are not publicly available due to ethical considerations.

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
