# Peer review of "Evaluating a Novel Extended Scope of Occupational Therapy Service Aimed at Hospital Avoidance in Tasmania, Australia, from the Perspective of Stakeholders"

_healthcare, 2022, doi:10.3390/healthcare10050842_

Round 1

Author Response

Dear Reviewer,

Please see the attachment responding to your comments.

Kind regards,

Pieter Van Dam

Reviewer 2 Report

Overall well written and interesting. I thought it was well presented and clear. It is good to see these sorts of service improvement activities subject to such a comprehensive analysis.

The first paragraph of the discussion sums up the aims and overall findings very well.

I have only a few points:

Major

I have some difficulty with the premise that addressing a risk would necessarily prevent an admission. While older people are much more likely to be admitted, it is unclear that this would prevent admissions as this is based on clinician view in this article.

Some sort of longitudinal data analysis using a difference in difference or similar approach might be able to prove causation but this is not provided here.

I would therefore like the authors to be more cautious with interpretation. You can argue that imminent risks were addressed which is expected to reduce avoidable hospitalisation however you also need to acknowledge the limitations of your analysis which cannot show causation.

Minor

Tables need to stand alone so provide explanation of acronyms in table notes. (e.g. MST FROP-COM) even if previously explained

Left align first column

Author Response

Dear Reviewer,

Please see the attachment responding  to your comments.

Kind regards,

Pieter

Round 2

Reviewer 1 Report

I have checked the revised manuscript and authors' responses. The manuscript has been significantly improved and responses are acceptable. I have no further comments.

Author Response

Dear reviewer,

Thank you for your time to review our manuscript for a second time and it is much appreciated.  I am pleased to read that our responses are acceptable and  that no further changes are required.

Kind regards,

Pieter Van Dam